# Assessment of the Clinical and Functional Health Status of Patients with Amyotrophic Lateral Sclerosis during the COVID-19 Pandemic in Brazil Using Telemedicine

**DOI:** 10.3390/healthcare12060627

**Published:** 2024-03-11

**Authors:** Ozana Brito, Guilherme Fregonezi, Karen Pondofe, Rayane Grayce da Silva Vieira, Tatiana Ribeiro, Mário Emílio Dourado Júnior, Emanuela Coriolano Fidelix, Danilo Nagem, Ricardo Valentim, Antonio Sarmento, Vanessa Resqueti

**Affiliations:** 1PneumoCardioVascular Lab/HUOL, Onofre Lopes University Hospital, Brazilian Hospital Services Company (EBSERH), Federal University of Rio Grande do Norte, Natal 59010-090, Brazil; ozana.brito.085@ufrn.edu.br (O.B.); fregonezi.guilherme@gmail.com (G.F.); karenpondofe@gmail.com (K.P.); rayane.vieira.071@ufrn.edu.br (R.G.d.S.V.); antonio_sarmento_@hotmail.com (A.S.); 2Laboratory of Technological Innovation in Health, Onofre Lopes University Hospital, Brazilian Hospital Services Company (EBSERH), Federal University of Rio Grande do Norte, Natal 59010-090, Brazil; ribeirotatiana.ufrn@gmail.com (T.R.); danilo.nagem@lais.huol.ufrn.br (D.N.); ricardo.valentim@lais.huol.ufrn.br (R.V.); 3Department of Physiotherapy, Onofre Lopes University Hospital, Brazilian Hospital Services Company (EBSERH), Federal University of Rio Grande do Norte, Natal 59078-970, Brazil; 4Laboratory of Intervention and Movement Analysis, Onofre Lopes University Hospital, Brazilian Hospital Services Company (EBSERH), Federal University of Rio Grande do Norte, Natal 59078-970, Brazil; 5Department of Integrated Medicine, Onofre Lopes University Hospital, Brazilian Hospital Services Company (EBSERH), Federal University of Rio Grande do Norte, Natal 59010-090, Brazil; medourado@ufrnet.br; 6Department of Neurology, Onofre Lopes University Hospital, Brazilian Hospital Services Company (EBSERH), Federal University of Rio Grande do Norte, Natal 59056-000, Brazil; efidelix@gmail.com; 7Department of Biomedical Engineering, Federal University of Rio Grande do Norte, Natal 59078-970, Brazil

**Keywords:** amyotrophic lateral sclerosis, COVID-19, telemedicine, non-invasive ventilation, functionality

## Abstract

This study aimed to monitor the clinical and functional progression of patients with amyotrophic lateral sclerosis (ALS) and adjust ventilatory support during the COVID-19 pandemic in Brazil using telemedicine. This longitudinal case series included five evaluations from January 2019 to June 2021. The first and second assessments were performed in person and consisted of pulmonary function, respiratory muscle strength, functionality (ALS Functional Rating Scale—Revised [ALSFRS-R]) and disease staging (King’s College criteria). The use of non-invasive ventilation (NIV), ALSFRS-R, and disease staging were assessed in the third, fourth, and fifth assessments during the COVID-19 pandemic, using telemedicine. The rate of functional decline was calculated by the difference in the total score of ALSFRS-R between evaluations. A cutoff of 0.77 in the ALSFRS-R was used to characterize the speed of functional decline. Eleven patients (mean age of 51 years, eight males) were assessed. The total score of the ALSFRS-R (*p* < 0.01) and its motor domain (*p* < 0.01) reduced significantly during the pandemic. NIV prescription increased from 54.4% to 83.3%. Telemedicine helped with the clinical and functional follow-up of patients with ALS.

## 1. Introduction

Amyotrophic lateral sclerosis (ALS) is marked by a gradual decline in functional capacity and increased morbidity from respiratory infections due to a progressive loss of muscle strength [1,2,3,4,5]. Limitations caused by these events differ among patients with spinal and bulbar onset according to disease phenotype [6,7]. Both groups of patients may also experience nocturnal hypoventilation and obstructive sleep apnea, increasing the burden of disease [8]. Alternatively, non-invasive ventilation (NIV) might improve quality of life and increase survival [8,9].

In this context, regular monitoring by a multidisciplinary team, adjustments to ventilatory parameters, and NIV maintenance are essential for disease management [2,8,9]. However, the COVID-19 pandemic and social isolation affected the care routine and impaired follow-up care. Health services needed to adapt to communication technologies (i.e., telemedicine) [10], which became the most viable option to monitor patients, address needs, and avoid contamination during the pandemic.

Nevertheless, questions and concerns related to telemedicine (e.g., use of instruments, clinical adherence, adjustment and prescription of ventilatory support, and whether telemedicine would identify the clinical and functional progression of patients with ALS) still emerged during the COVID-19 pandemic [11,12]. We hypothesized that patients with ALS experience rapid and significant disease progression due to the clinical nature of the disease. During the COVID-19 pandemic, people found themselves isolated and experiencing difficulties in accessing health care, which could lead to a worsening of their illnesses. This study aimed to monitor the clinical and functional evolution of ALS patients during the COVID-19 pandemic (April 2020 to June 2021) using telemedicine assistance.

## 2. Materials and Methods

### 2.1. Study Type and Participants

This longitudinal case series was developed at the Pneumocardiovascular Laboratory of Onofre Lopes University Hospital (HUOL/EBSERH). Patients with bulbar or spinal ALS of both sexes, aged over 18 years, classified according to the El Escorial World Federation of Neurology criteria [13], that were included before the COVID-19 pandemic with two clinical assessments, and continued follow-up by a multidisciplinary team (neurologist, physiotherapist, psychologist, speech therapist, and nutritionist) were included. Those without internet access or who could not respond to questionnaires with or without the help of caregivers (including patients with limited communication or using alternative communication tools) were excluded. Strobe Checklist was applied to verify essential items for the research (Appendix A). All study procedures were approved by the research ethics committee of the HUOL/EBSERH (3.735.479/2019) and followed the Declaration of Helsinki and resolution 466/12 of the National Health Council.

### 2.2. Patient and Public Involvement

A patient and public involvement panel was not recruited to inform the design, recruitment, conduct, or dissemination plan for this study.

### 2.3. Study Design

The follow-up occurred from January 2019 to June 2021. Five assessments were performed: A1, A2, A3, A4, and A5. Assessments A1 and A2 occurred in person before the COVID-19 pandemic (January 2019 to March 2020), whereas A3 to A5 were performed remotely during the COVID-19 pandemic (April 2020 to June 2021).

The first assessments (A1 and A2) included pulmonary function, respiratory muscle strength, functionality (ALS Functional Rating Scale—Revised [ALSFRS-R/BR]) [14], rate of disease progression, and disease staging (King’s College criteria) [15]. Further assessments (A3 to A5) included a clinical evaluation form, use of NIV, assessment of NIV parameters, ALSFRS-R/BR, rate of disease progression, and King’s College criteria (Figure 1).

For online evaluations (A3 to A5), we used telecommunication platforms (e.g., Google Meet^®^ (Google Corp, Mountain View, CA, USA, 2017), WhatsApp^®^ (WhatsApp, Menlo Park, CA, USA, 2009), or the Rio Grande do Norte Teleconsultation System). The Rio Grande do Norte Teleconsultation System is a platform for remote communication between health professionals and patients, developed in a partnership between the Ministry of Health and the Laboratory of Technological Innovation in Health at the Federal University of Rio Grande do Norte (LAIS-UFRN) [16]. The platform is part of the Telehealth Brazil Networks Program, a national-level action that seeks to provide better quality care for professionals and basic health care in the Unified Health System, especially during the social isolation caused by the COVID-19 pandemic. Its operation is regulated through Ordinance No. 467 of 20 March 2020 from the Ministry of Health’s Cabinet [17].

The team identified the need for NIV prescription or adjustments according to demands found during assessments A3 to A5. The monitoring of disease progression and guidance were conducted during synchronous (real-time via Google Meet^®^ or Rio Grande do Norte Teleconsultation System platforms) or asynchronous assessments (via WhatsApp^®^) to complement the information from synchronous consultations.

### 2.4. Pulmonary Function and Respiratory Muscle Strength

Pulmonary function and respiratory muscle strength were assessed in person only before the pandemic (assessments A1 and A2).

Pulmonary function was assessed via spirometry using a Koko DigiDoser spirometer (Longmont, CO, USA), according to the American Thoracic Society/European Respiratory Society [18]. Forced vital capacity (L), forced expiratory volume in the first second (L), forced expiratory flow at 25–75% (L/s), and peak expiratory flow (L/s) were considered in absolute and predicted values [19].

The following inspiratory and expiratory muscle strength parameters were assessed using a digital manovacuometer (NEPEB-LabCare/UFMG, Belo Horizonte, MG, Brazil): maximal inspiratory pressure (cmH_2_O), maximal expiratory pressure (cmH_2_O), and sniff nasal inspiratory pressure (cmH_2_O). Predicted values followed those recommended for the Brazilian population [20,21].

### 2.5. Functionality and Rate of Disease Progression

Functionality was assessed using the ALSFRS-R, adapted for Brazilian Portuguese [14]. The scale consists of 12 questions related to daily activities and symptoms, divided into four domains: bulbar, upper limb, lower limb, and respiratory. Domain scores range from 0 to 4; the maximum score is 48 points (i.e., normal functionality). The same previously trained researcher administered the ALSFRS-R in all assessments.

Scores of the upper and lower limb domains were summed and represented the motor domain (maximum score of 8 points). The rate of disease progression was calculated by subtracting ALSFRS-R total scores between assessments and dividing by time (in months) between assessments [12]. We considered intervals between A1–A2 and A3–A5 to calculate the rate of disease progression before and during the pandemic, respectively. A cutoff point of 0.77 was used to represent the slow (<0.77) and fast (>0.77) rate of disease progression [22].

### 2.6. Disease Staging

We used the King’s College [15] criteria to classify disease staging retrospectively according to ALSFRS-R scores and functional impairments [23] (Figure 2). It categorizes patients into five stages, according to clinical milestones: (1) functional involvement of one central nervous system area (symptom onset), (2) functional involvement of two central nervous system areas, (3) functional involvement of three central nervous system areas, (4a) need for gastrostomy or (4b) NIV, and (5) death.

### 2.7. Clinical Evaluation and Use of NIV

Clinical evaluation was performed using a form developed by the researchers and adapted to A3, A4, and A5 assessments. The form included self-reported questions (“yes” or “no”) about signs and symptoms of respiratory infections, nocturnal hypoventilation (nocturnal awakenings, morning headaches, and daytime drowsiness), use of NIV (adherence, time, and type of interface), and NIV adjustments.

### 2.8. Adjustment and Prescription of NIV Support

NIV prescription was based on the ALSFRS-R respiratory domain, signs and symptoms of nocturnal hypoventilation, and use of NIV. An experienced neurologist from the multidisciplinary team conducted the prescription in consensus with other professionals.

For patients already using ventilatory support, the parameters were adjusted according to the clinical and functional evaluation or by caregiver request. One professional instructed the adjustments with the local team or caregiver remotely, using telemedicine. Some NIV equipment allowed for the adjusting of parameters using an online application.

### 2.9. Statistical Analysis

Statistical analyses were performed using GraphPad Prism software (version 8.0, La Jolla, CA, USA) and the Statistical Package for the Social Sciences (version 20.0, IBM Corp, Armonk, NY, USA). Descriptive analysis was performed using the median and interquartile range (25% and 75%). The Shapiro–Wilk test verified data normality. For intergroup analyses, the sample was divided into two subgroups according to the clinical diagnosis of ALS (spinal or bulbar). For intragroup analyses, comparisons between assessments (A1, A2, A3, A4, A5) were considered. Functional impairment was quantified by considering a significant decrease in the total and domain scores of the ALSFRS-R. Intra- and intergroup data (spinal and bulbar) were compared using Friedman (Dunn’s post hoc) and Mann–Whitney tests, respectively. For categorical variables, the Chi-squared test was used for analyses with more than two categories, and Fisher’s exact test for dichotomous categorical variables.

## 3. Results

We followed 11 patients for 30 months (January 2019 to June 2021). Most patients had spinal onset ALS (72.7%), were male (72.7%), had a median age of 51 years (interquartile range 43 to 55), and resided in inland cities of the Rio Grande do Norte state (72.7%) far from the multidisciplinary care center. Table 1 shows anthropometric and pulmonary function data. Baseline data regarding FVC [%Pred], MEP [cmH_2_O], SNIP [cmH_2_O], and SNIP [%Pred] between Spina ALS and Bulbar ALS showed no significant difference between them.

Total ALSFRS-R (*p* < 0.01) and motor domain (*p* < 0.01) scores were reduced during the pandemic. However, total and domain scores were not different between patients with spinal and bulbar onset (Figure 3). According to ALSFRS-R, 90.9% of patients presented a slow disease progression between A1–A2 and 72.7% between A3–A5 assessments; no statistical difference was observed when comparing the ALSFRS total score between all assessments (*p* = 0.29).

A total of 45.4% and 72.7% of patients were classified as stage 4 at A1 and A5, respectively; however, no statistical significance was found between A1 and A5. Dyspnea (A3 = 45.4%, A4 = 45.4%, and A5 = 36.3%) (Table 2) and nocturnal awakening (A3 = 54.5%, A4 = 63.3%, and A5 = 27.2%) were the most reported signs of nocturnal hypoventilation. No significant difference was observed between moments (A3, A4, and A5).

Most patients used NIV at night and during a half-day shift (A3 = 60%, A4 = 16.6%, and A5 = 50%) (Figure 4). The NIV of two patients was adjusted during A4 since they reported increased respiratory effort, need for ventilation during the day, or because NIV parameters no longer met demands. After adjustments, one patient reported decreased respiratory effort and a decrease in the frequency of NIV use during the day. Another patient reported reduced dyspnea; however, NIV was required during the day. We observed that all patients adhered to treatment during A5, mainly due to remote guidance on NIV.

## 4. Discussion

This study aimed to monitor the clinical and functional progression of patients with ALS during the COVID-19 pandemic in Brazil using telemedicine. The main findings were that (1) functional capacity reduced with disease progression and (2) telemedicine proved successful for remote clinical management since it allowed for identifying signs and symptoms and facilitated the prescription and adjustment of NIV during social isolation.

A gradual loss of functional capacity is expected in patients with ALS due to progressive muscle weakness [24]. Our study found a significant functional decrease and a slow rate of disease progression using remote assessments during the COVID-19 pandemic. Gonçalves et al. [12] observed similar results in 32 patients who interrupted their multidisciplinary care routine due to the pandemic. However, in that study [12], the follow-up was conducted during four months at the beginning of the pandemic, and patients showed a fast rate of disease progression, whereas we assessed patients before the social isolation period and monitored them regularly during 14 months of the pandemic. Both studies demonstrated that remote monitoring may help monitor the rate of disease progression in this population.

The clinical progression of ALS varies among patients. Thus, a system that allows for standardized measurements and is easy to understand by professionals is important for managing resources and care [25,26]. An important aspect of our study was that disease staging was classified remotely using the King’s College criteria. To our knowledge, only Manera et al. [27] classified patients with ALS using telemedicine and observed a high reliability of this system. Van Eijck et al. also demonstrated good correlations between disease staging, assessed remotely using King’s College criteria, and other objective measures of clinical disease progression [28]. These results support ours, indicating that telemedicine allows for the follow-up of patients with ALS and establishing King’s College criteria as a valuable tool for this purpose.

Another important aspect of this study was the remote monitoring of signs and symptoms of nocturnal hypoventilation. Early monitoring increases the survival of patients with ALS [29] and is relevant because nocturnal hypoventilation reflects respiratory muscle impairment and the need for ventilatory support [8,9]. In our study, signs of nocturnal hypoventilation and questions of the ALSFRS-R respiratory domain helped prescribe ventilatory support. During social isolation, only Marchi et al. reported NIV prescriptions for patients with ALS [30]. However, this prescription was requested after a telemedicine evaluation and conducted in person by a pulmonologist who did not participate in the study. Although we remotely prescribed and adjusted the NIV, results from the study of Marchi et al. support the importance of remote care provided by a multidisciplinary team since patients remained stable without exacerbations. Our findings demonstrate that remote assessment and direct contact with the multidisciplinary team may have also allowed for the early identification of symptoms of respiratory exacerbation and favored remote decision-making and successful NIV prescription for this population. Furthermore, all patients adhered to NIV use at the end of the study.

As study limitations, a small sample size was achieved since patients without prior assessment in our database were excluded. Invasive measurements and in-person assessments during the COVID-19 pandemic (respiratory muscle strength and pulmonary function) were also not performed.

## 5. Conclusions

Telemedicine facilitated the monitoring of the clinical and functional progression of patients with ALS during the COVID-19 pandemic. With a few adjustments, it was possible to reduce the rate of disease progression and maintain functionality between different moments. NIV adjustments were important in reducing symptoms such as dyspnea and patient adherence to ventilatory equipment.

Patients had good adherence to telemedicine, with a feeling of safety and support from professionals during the COVID-19 isolation period. It was possible to monitor and care for ALS patients through telemedicine. However, due to the rare nature of the disease, its rapid progression, and some limitations that telemedicine brings, such as preventing in-person assessments, the results found become somewhat restricted.

We suggest that the King’s College criteria and a subjective assessment of sleep through telemedicine be used in the care of patients with ALS.

## Figures and Tables

**Figure 1 healthcare-12-00627-f001:**
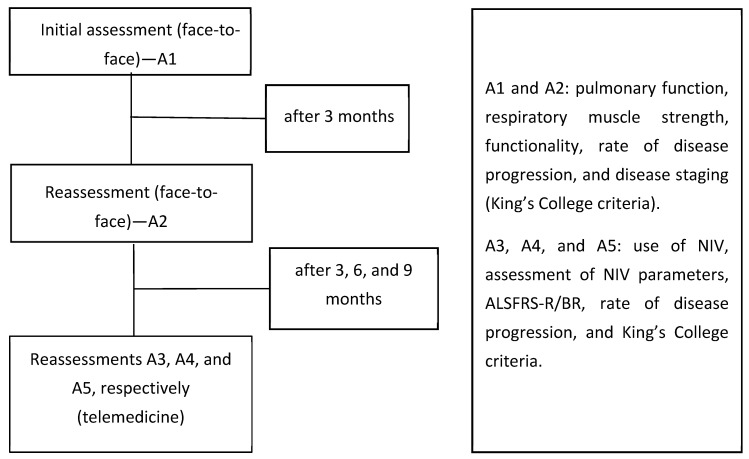
Study design flowchart.

**Figure 2 healthcare-12-00627-f002:**
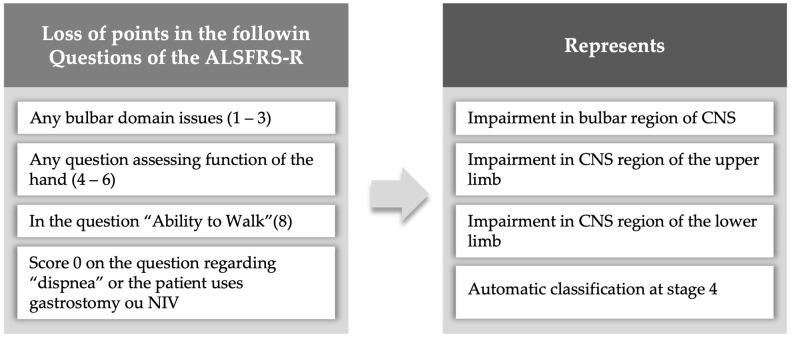
Staging criteria of the King’s College system according to ALSFRS-R score. CNS: central nervous system; NIV: non-invasive ventilation.

**Figure 3 healthcare-12-00627-f003:**
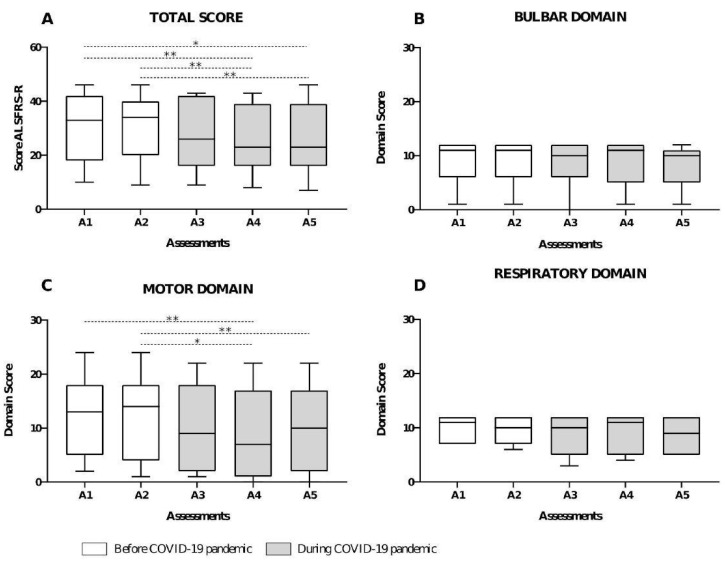
Disease progression according to ALSFRS-R. (**A**) ALSFRS-R total score at each assessment moment. (**B**) Bulbar domain score in each assessment. (**C**) Motor domain score in each assessment. (**D**) Respiratory domain score in each assessment. * *p* < 0.05, ** *p* < 0.01.

**Figure 4 healthcare-12-00627-f004:**
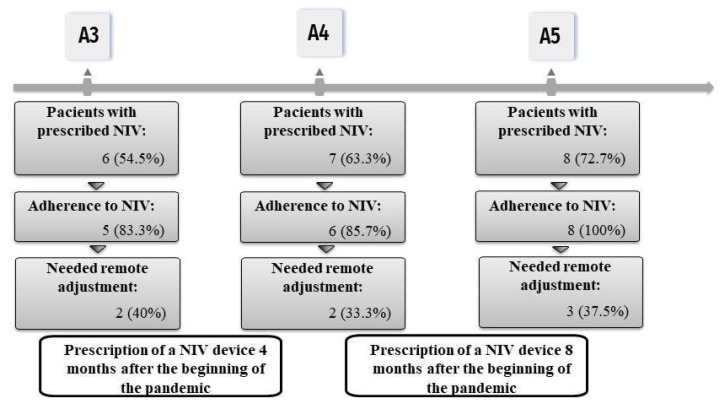
Non-invasive ventilatory support during the pandemic. Relative values of NIV use, adherence to use and parameter adjustments made during assessments A3, A4 and A5.

**Table 1 healthcare-12-00627-t001:** Sample characterization (*n* = 11) at baseline.

Variables	All Patients (*n* = 11)	Spinal ALS (*n* = 8)	Bulbar ALS (*n* = 3)
Sex			
Male	8 (72.7%)	6 (75%)	2 (66.6%)
Female	3 (27.7%)	2 (25%)	1 (33.3%)
Age [years]	51 (43–55)	47.5 (41.5–51.7)	64 (55–73)
El Escorial			
Possible	-	-	-
Probable	3 (27%)	2 (25%)	1 (50%)
Confirmed	8 (72.7%)	6 (75%)	1 (50%)
Origin			
State capital	3 (27.7%)	2 (22.2%)	1 (50%)
Inland cities	8 (72.2%)	7 (77.7%)	1 (50%)
FVC [L]	3.1 (2.1–3.9)	3.4 (2.7–4.4)	2.3 (1.8–2.6)
FVC [%Pred]	69.6 (56.5–96.7)	69.6 (60–97.8)	65.9 (41.1–90.8)
FEV_1_ [L]	2.3 (1.6–3.2)	2.8 (2–3.5)	1.6 (1.6–2.2)
FEV_1_ [%Pred]	74 (55.8–89.2)	81.2 (63.9–92.2)	65.4 (44.1–80.2)
FEF_25–75%_ [L/s]	2.7 (1.5–3.5)	3.4 (1.5–3.6)	1.7 (1.3–2.7)
FEF_25–75%_ [%Pred]	68.4 (56.8–90.9)	77.4 (58.6–96.2)	68.4 (49.5–84)
PEF [L/s]	6.0 (1.7–6.6)	6.1 (3.5–6.7)	1.9 (1.6–6.4)
MIP [cmH_2_O]	67 (27.7–86.5)	54.5 (24.7–99.2)	69 (30.4–82)
MIP [%Pred]	60 (28.9–81.7)	46.4 (28.1–84.3)	67.2 (27.8–92.3)
MEP [cmH_2_O]	59 (35.5–81.5)	60.5 (40.2–98.7)	59 (25–85)
MEP [%Pred]	59 (42.2–70.4)	62.2 (44.8–85.1)	52.4 (19.9–69.9)
SNIP [cmH_2_O]	72 (44–79.5)	74 (39.2–87.2)	68 (46–72)
SNIP [%Pred]	64.5 (42.6–85.9)	65.3 (41.7–92.8)	61.9 (40.8–85.8)

Data presented as absolute (relative) frequencies and median (interquartile range 25–75%). FVC: forced vital capacity; FEV_1_: forced expiratory volume in the first second; FEF_25–75%_: forced expiratory flow 25–75%; MIP: maximal inspiratory pressure; MEP: maximal expiratory pressure; SNIP: sniff nasal inspiratory pressure; L/s: liters per second; cmH_2_O: centimeters of water; %Pred: percentage of predicted.

**Table 2 healthcare-12-00627-t002:** Respiratory symptoms during the pandemic (*n* = 11).

Variables	A3	A4	A5
Dyspnea	5 (45.4%)	5 (45.4%)	4 (36.3%)
Fatigue	5 (45.4%)	8 (72.7%)	6 (54.4%)
Fever	2 (25%)	1 (9.0%)	-
Sore throat	-	1 (9.0%)	2 (18.1%)
Runny nose	2 (25%)	3 (27.2%)	2 (18.1%)
Nasal Obstruction	4 (50%)	6 (54.4%)	5 (45.%)
Cough			
Effective	6 (54.5%)	5 (45.4%)	5 (45.4%)
Ineffective	5 (45.4%)	6 (54.5%)	6 (54.5%)

Data presented as absolute (relative) frequencies. A3: third assessment; A4: fourth assessment; and A5: fifth assessment.

## Data Availability

All data relevant to the understanding of this study are included in the article or uploaded as Appendix A.

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
