# Peer review of "Assessment of the Clinical and Functional Health Status of Patients with Amyotrophic Lateral Sclerosis during the COVID-19 Pandemic in Brazil Using Telemedicine"

_healthcare, 2024, doi:10.3390/healthcare12060627_

Round 1

Reviewer 1 Report

Comments and Suggestions for Authors

Authors conduct the assessment of ALS patients using telemedicine during COVID-19 pandemic in Brazil. This manuscript has some value to be published in Healthcare”, but there are some specific comments. 

Comments to the authors.  

Major comments;   

Authors follow 11 patients for 30 months (January 2019 to June 2021) and monitor the patients’ clinical condition. There are no study/results of “Patients own assessment about this telemedicine”.

P8, L239; “Consequently, the long-term monitoring of our study may have influenced in the difference the rate of disease progression observed between studies and this suggested remote monitoring can help to reduce the speed of disease progression”. This conclusion is not appropriate, because there is no control cohort (patients’ group without telemedicine) in this study.

Minor comments;

It is not described the detail of “a multidisciplinary team”.

P3, L119; Monitoring of disease progression and guidance were also performed during synchronous (communication performed in real-time) or asynchronous (communication without face-to-face contact) assessments. How many patients are in real-time or non face-to-face? Are there any difference in the monitoring results?

Author Response

Authors follow 11 patients for 30 months (January 2019 to June 2021) and monitor the patients’ clinical condition. There are no study/results of “Patients own assessment about this telemedicine”.

Response; There was no evaluation through a questionnaire of patient satisfaction in relation to telemedicine, but during the care, caregivers and/or patients thanked them for the follow-up/monitoring and guidance given as patients were left without in-person assistance during the pandemic.

P8, L239; “Consequently, the long-term monitoring of our study may have influenced in the difference the rate of disease progression observed between studies and this suggested remote monitoring can help to reduce the speed of disease progression”. This conclusion is not appropriate, because there is no control cohort (patients’ group without telemedicine) in this study.

Response; We agree with the comment and thank you. The conclusion was withdrawn.

It is not described the detail of “a multidisciplinary team”.

Response; We specify the members of the multidisciplinary team in the material and methods topic, P2, L86-88; “Patients with bulbar or spinal ALS of both sexes, aged over 18 years, classified according to the El Escorial World Federation of Neurology criteria [13], under follow-up by a multidisciplinary team (neurologist, physiotherapist, psychologist, speech therapist, nutritionist) before the COVID-19 pandemic were included.”

P3, L119; Monitoring of disease progression and guidance were also performed during synchronous (communication performed in real-time) or asynchronous (communication without face-to-face contact) assessments. How many patients are in real-time or non face-to-face? Are there any difference in the monitoring results?

Response; We complement the information as follows; P3, 120-124; Monitoring of disease progression and guidance were also carried out during synchronous assessments (communication carried out in real time via the Google Meet® or Rio Grande do Norte Teleconsultation System platforms) or asynchronous assessments (communication via Whatsapp®) to complement the information from synchronous consultations. and changing materials for A3 to A5.

Reviewer 2 Report

Comments and Suggestions for Authors

The authors provided the results of the report on the clinical and functional progression (improvement of the health status?) of patients with amyotrophic lateral sclerosis (ALS) and adjust(ing?) ventilatory support during the COVID-19 pandemic in Brazil using telemedicine. They evaluated in the patients the pulmonary function, the respiratory muscle strength, provided the data on functionality (ALS Functional Rating Scale-Revised [ALS-FRS-R]), and disease staging (King’s College? criteria). The use of non-invasive ventilation (NIV), ALSFRS-R, and disease staging were assessed in the third, fourth, and fifth assessments during the COVID-19 pandemic using telemedicine. The concluded that prescription of NIV increased from 54.4% to 83.3% as well as the monitored parameters improved. Telemedicine helped clinical and functional follow-up of patients with ALS.

The scientific and cognitive value is low and the results are obvious and easy to predict.

The manuscript is full of grammatical errors in English, the message of which is poorly understood in many parts of the text. 

The title is confusing and would be better understood as follows: Assessment of clinical and functional health status of patients with Amyotrophic Lateral Sclerosis during COVID-19 pandemic in Brazil using telemedicine: a longitudinal case series study.

The manuscript is a case report.

In the Introduction, the state-of-art is presented superficially and unconvincingly. Although the authors selected the literature well, they did not briefly present their content in the Introduction.

The list of references is editorially inconsistent with that adopted in MDPI.

The intentions of the authors in the sentence in lines 94-96 (2.2. Patient and Public Involvement) are confusing. They should provide the anthropometric and demographic data on the patients’ group and the number of patients.

It is not clear, if and how all patients have performed the clinical studies and these described in 2.4. Pulmonary function and respiratory muscle strength.

The description of the results in the text is insufficient.

The graphical regime in Figures 3-5 is different.

The conclusions are very general.

The Materials and Methods section, especially for the study design description might be clarified with the flow chart.

Comments on the Quality of English Language

English needs significant improvements.

Author Response

- The scientific and cognitive value is low and the results are obvious and easy to predict.

Response; Thanks for the comment. We have been working with patients with ALS since 2016 and we always try to find some way to help the patients and family members involved as it is a rapidly progressing disease with few treatment possibilities to improve symptoms. The pandemic was a drastic period for our patients, who were isolated. We are happy to have been able to provide assistance, even if minimal, through telemedicine, with results equivalent to outpatient care.

- The manuscript is full of grammatical errors in English, the message of which is poorly understood in many parts of the text. 

Response; We regret the errors. We hired @provatis.academy to correct grammar.

- The title is confusing and would be better understood as follows: Assessment of clinical and functional health status of patients with Amyotrophic Lateral Sclerosis during COVID-19 pandemic in Brazil using telemedicine: case report.

Response; Thank you for the suggestion and we have changed the title.

- The list of references is editorially inconsistent with that adopted in MDPI.

Response; Thanks for the observation. We corrected the references according to adoption in MDPI.

- The intentions of the authors in the sentence in lines 94-96 (2.2. Patient and Public Involvement) are confusing. They should provide the anthropometric and demographic data on the patients’ group and the number of patients.

Response; The text has been modified for better understanding. A patient and public involvement panel’ was not specifically recruited to inform the design, recruitment, conduct or dissemination plan for this study.

- It is not clear, if and how all patients have performed the clinical studies and these described in 2.4. Pulmonary function and respiratory muscle strength.

The volunteers were assessed for pulmonary function and respiratory muscle strength before the pandemic, using the most recent values.

English needs significant improvements.

The text has been adjusted and corrected for English.

Round 2

Reviewer 1 Report

Comments and Suggestions for Authors

This revised manuscript is written correctly according to the reviewers' comments.

P9, L280; yhe > the   

Author Response

Dear Reviewer 1.

We appreciate all the comments and are happy to be able to resolve the issues.

Reviewer 2 Report

Comments and Suggestions for Authors

Comparing the versions of the manuscripts provided for the first and the second reviews may lead to the conclusions as follows:

The Authors answered more than half of my suggestions and queries well.

The expression of the English and the grammar rules improved a bit.

To my query:

"- The title is confusing and would be better understood as follows: Assessment of clinical and functional health status of patients with Amyotrophic Lateral Sclerosis during COVID-19 pandemic in Brazil using telemedicine."

the authors wrote in their response:

"Response; Thank you for the suggestion and we have changed the title."

but I can’t see any change in the two manuscripts.

They did not answer to the following suggestions:

…“The description of the results in the text is insufficient.

The graphical regime in Figures 3-5 is different.

The conclusions are very general.

The Materials and Methods section, especially for the study design description might be clarified with the flow chart….

I consider the mentioned as important and I recommend them as minor revisions.

Comments on the Quality of English Language

Minor corrections are required including editorial.

Author Response

Dear Reviewer 2.

Thank you for posting the improvements.

We corrected the title error and changed it to: "Assessment of clinical and functional health status of patients with Amyotrophic Lateral Sclerosis during COVID-19 pandemic in Brazil using telemedicine", according to your suggestion.

As for English, we requested new corrections from PROVATIS Academy. We believe that the agreement errors have been corrected.

We included the flowchart figure for better clarification in the study design.

In the results session, we corrected some data, described it objectively and removed figures 4 and 5, as they were not adding significant data to the text.

The conclusions were reconstructed with descriptions of final results that respond to the objective of the research in terms of monitoring individuals with it through telemedicine.

We hope that the corrections were sufficient and we are available to resolve any doubts.